# Tritiation of aryl thianthrenium salts with a molecular palladium catalyst

Da Zhao[1], Roland Petzold[1], Jiyao Yan[1,2], Dieter Muri[3] & Tobias Ritter[1✉]

Tritium labelling is a critical tool for investigating the pharmacokinetic and pharmacodynamic properties of drugs, autoradiography, receptor binding and receptor occupancy studies[1]. Tritium gas is the preferred source of tritium for the preparation of labelled molecules because it is available in high isotopic purity[2]. The introduction of tritium labels from tritium gas is commonly achieved by heterogeneous transition-metal-catalysed tritiation of aryl (pseudo)halides. However, heterogeneous catalysts such as palladium supported on carbon operate through a reaction mechanism that also results in the reduction of other functional groups that are prominently featured in pharmaceuticals[3]. Homogeneous palladium catalysts can react chemoselectively with aryl (pseudo)halides but have not been used for hydrogenolysis reactions because, after required oxidative addition, they cannot split dihydrogen[4]. Here we report a homogenous hydrogenolysis reaction with a well defined, molecular palladium catalyst. We show how the thianthrene leaving group—which can be introduced selectively into pharmaceuticals by late-stage C–H functionalization[5]—differs in its coordinating ability to relevant palladium(II) catalysts from conventional leaving groups to enable the previously unrealized catalysis with dihydrogen. This distinct reactivity combined with the chemoselectivity of a well defined molecular palladium catalyst enables the tritiation of small-molecule pharmaceuticals that contain functionality that may otherwise not be tolerated by heterogeneous catalysts. The tritiation reaction does not require an inert atmosphere or dry conditions and is therefore practical and robust to execute, and could have an immediate impact in the discovery and development of pharmaceuticals.

Tritium ($^3$H) labelling allows the direct incorporation of a radioactive tag into pharmaceutical candidates without substantial changes in their chemical and physical properties and biological activity[6]. However, many reliable hydrogenation or hydrogenolysis reactions cannot be suitably used for tritium labelling owing to a lack of reagents, low molar activity, restricted functional-group tolerance or safety concerns. Tritiated water ($^3$H$_2$O) is problematic owing to fast washout of the label from omnipresent water and safety concerns regarding the potential fast uptake of radioactive water by experimentalists. The preferred source of tritium labels is tritium gas ($^3$H$_2$), which is available in high isotopic purity and practical to handle with commercially available manifolds on the small scale typically used for labelling[2,7].

Hydrogenation with hydrogen gas is one of the most extensively studied reactions in chemistry, with numerous important applications ranging from biomass degradation[8] to hydrogenolysis of otherwise persistent halogenated pollutants[9]. Several well defined, homogeneous transition-metal catalysts based on rhodium[10], iridium[11] and ruthenium[12] can split the strong hydrogen–hydrogen bond for countless productive hydrogenation reactions of unsaturated bonds. However, appropriate unsaturated bonds are often not present in pharmaceuticals or would

be destroyed by hydrogenation, and the same hydrogenation catalysts are generally not useful for the hydrogenolysis of carbon–halide bonds because most transition-metal hydrides are inactive towards the oxidative addition of carbon–heteroatom bonds. In the presence of both dihydrogen and aryl (pseudo)halides, dihydrogen oxidative addition is commonly faster, which results in metal hydrides in higher oxidation states that are not suitable for aryl (pseudo)halide oxidative addition[13,14]. Therefore, for the hydrogenolysis of carbon–heteroatom bonds[15], chemists select heterogeneous catalysts, such as palladium supported on carbon, which can effectively reduce aryl(pseudo)halides through a mechanistically distinct pathway[16,17]. The reactivity of the active hydrogen chemisorbed on the catalyst surface results in low chemoselectivity, and the undesired reduction of other functional groups typically found in pharmaceuticals (Fig. 1a)[3].

Tritium for hydrogen exchange is a desirable way for tritium incorporation because prior functionalization is not required[18]. Several impressive hydrogen isotope exchange reactions have been developed with transition-metal catalysts, maybe most notably those based on iridium[19,20] and nickel[21]; however, they require the presence of directing groups or heterocycles for efficient transformations.

[1]Max-Planck-Institut für Kohlenforschung, Mülheim an der Ruhr, Germany. [2]Institute of Organic Chemistry, RWTH Aachen University, Aachen, Germany. [3]Pre-clinical CMC, Roche Pharma Research and Early Development, Roche Innovation Center Basel, Basel, Switzerland. ✉e-mail: ritter@kofo.mpg.de

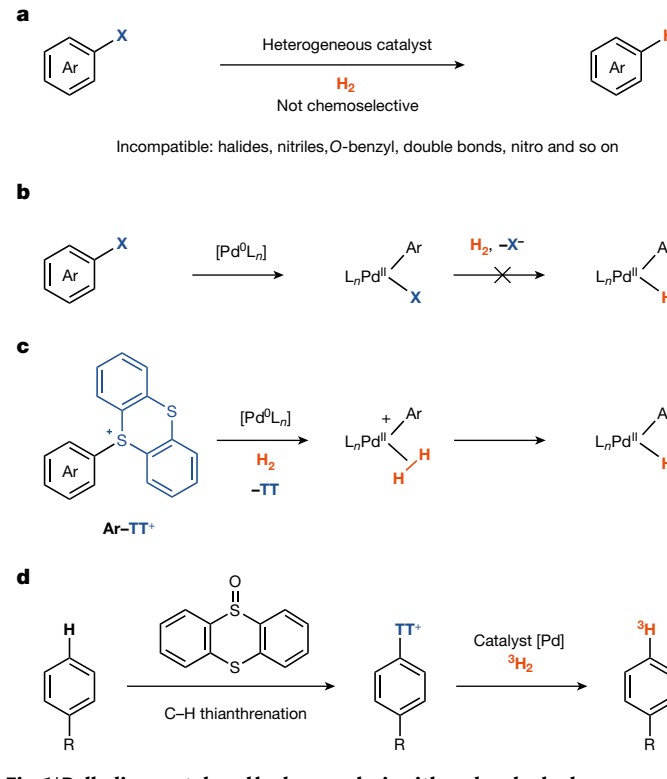

**Fig. 1 | Palladium-catalysed hydrogenolysis with molecular hydrogen.**
**a**, Heterogeneous Pd-catalysed hydrogenolysis with $H_2$. **b**, Homogeneous Pd-catalysed hydrogenolysis of aryl halide. L, neutral 2-electron ligand. **c**, Homogeneous Pd-catalysed hydrogenolysis of aryl thianthrenium salt. **d**, Chemo- and site-selective C–H tritiation via arylthianthrenium salt by homogeneous palladium catalysis. X, conventional (pseudo)halide.

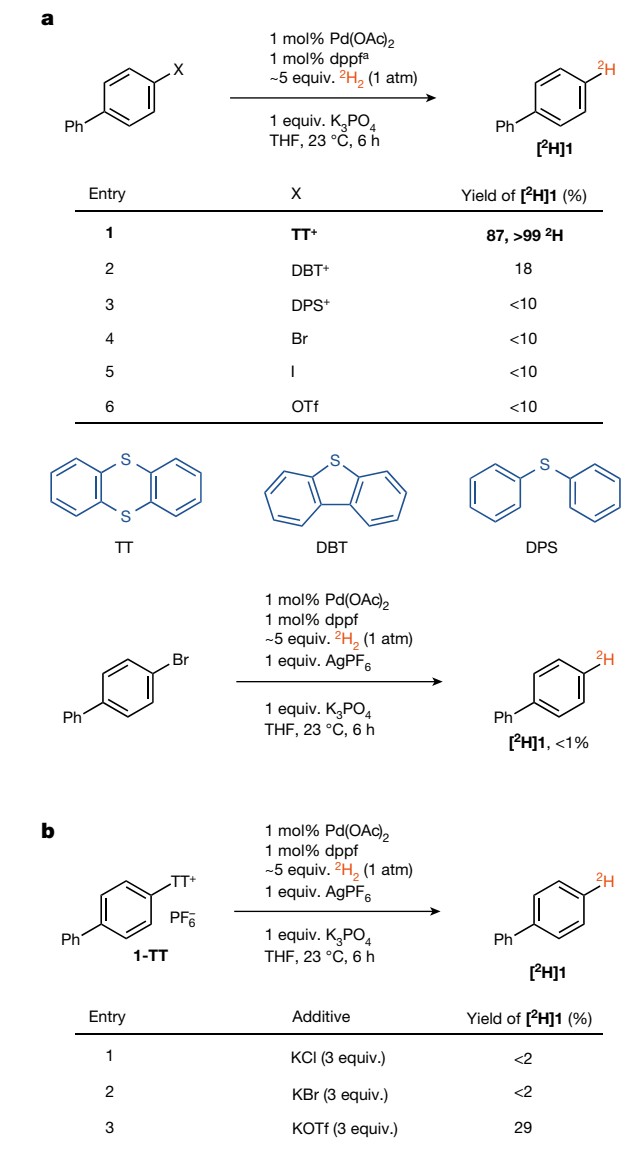

**Fig. 2 | Homogeneous palladium-catalysed reductive deuteration of aryl thianthrenium salt and aryl (pseudo)halides. a**, Hydrogenolysis of aryl (pseudo) halides. dppf, 1,1'-bis(diphenylphosphino)ferrocene. [a]For an additional evaluation of highly active monodentate phosphine ligands, see Supplementary Table 11. **b**, Effect of halide anions in hydrogenolysis of aryl thianthrenium salt.

An aromatic hydrogen isotope exchange reaction that does not require directing groups was developed based on an iron catalyst, which provides successful tritiation complementary to the iridium-catalysed methods[22]. At present, the low-valent iron catalyst does not tolerate protic functional groups and requires an inert atmosphere, which complicates its routine, practical application in pharmaceutical development[23,24]. In addition, all hydrogen isotope exchange reactions typically afford several constitutional isomers with more than one label incorporated, which can be advantageous because high molar activity can be reached[25], but may also render interpretation of imaging studies difficult when the labelled molecule is metabolized[26]. In that sense, a robust, practical, site-selective functionalization would be complementary to the hydrogen exchange reactions and could add significant value to the field[27].

Metal-catalysed hydrogenolysis proceeds through metal-hydride intermediates[28] that can be formed by dihydrogen oxidative addition. Transition-metal complexes that react with aryl (pseudo)halides through oxidative addition do not engage with dihydrogen in a second oxidative reaction to form metal hydrides. Heterolytic cleavage of the hydrogen–hydrogen bond can also result in the formation of metal hydrides[29]. For example, electrophilic transition-metal complexes of iridium[30] and ruthenium[31] can substantially acidify dihydrogen through complexation owing to a strong σ donor–acceptor interaction[29,32], which can result in the formation of metal hydrides through deprotonation of the metal $\eta^2$-dihydrogen complexes[33]. Only a few examples of palladium (Pd) dihydrogen complexes are known[4], and no electrophilic Pd(II) catalysts—generated through oxidative addition of an aryl (pseudo)halide to low-valent palladium—have been shown to heterolyse dihydrogen[34–36]. Upon oxidative addition to a ligated Pd(0), no appropriate coordination site is available at square planar $d^8$

Pd(II) tetracoordinate complexes for dihydrogen coordination because (pseudo)halides outcompete dihydrogen from coordination to Pd(II) (Fig. 1b). Aryl thianthrenium salts can be more reactive than aryl halides and aryl triflates in cross-coupling reactions[5,37] and photoredox catalysis[38–40], and they are readily accessed by direct, late-stage C–H functionalization of complex small-molecule arenes in exquisite site selectivity. The fundamental novelty reported here entails that dihydrogen can be split by a palladium catalyst as part of a productive catalytic cycle that includes oxidative addition of aryl thianthrenium salts (Fig. 1c). The ability to engage structurally complex arenes and small-molecule pharmaceuticals at a late stage without directing groups, excellent functional group tolerance, exquisite site selectivity and high isotopic purity, in combination with operational simplicity, distinguish this protocol from heterogeneous hydrodehalogenation and hydrogen isotope exchange reactions for practical and robust tritiation (Fig. 1d).

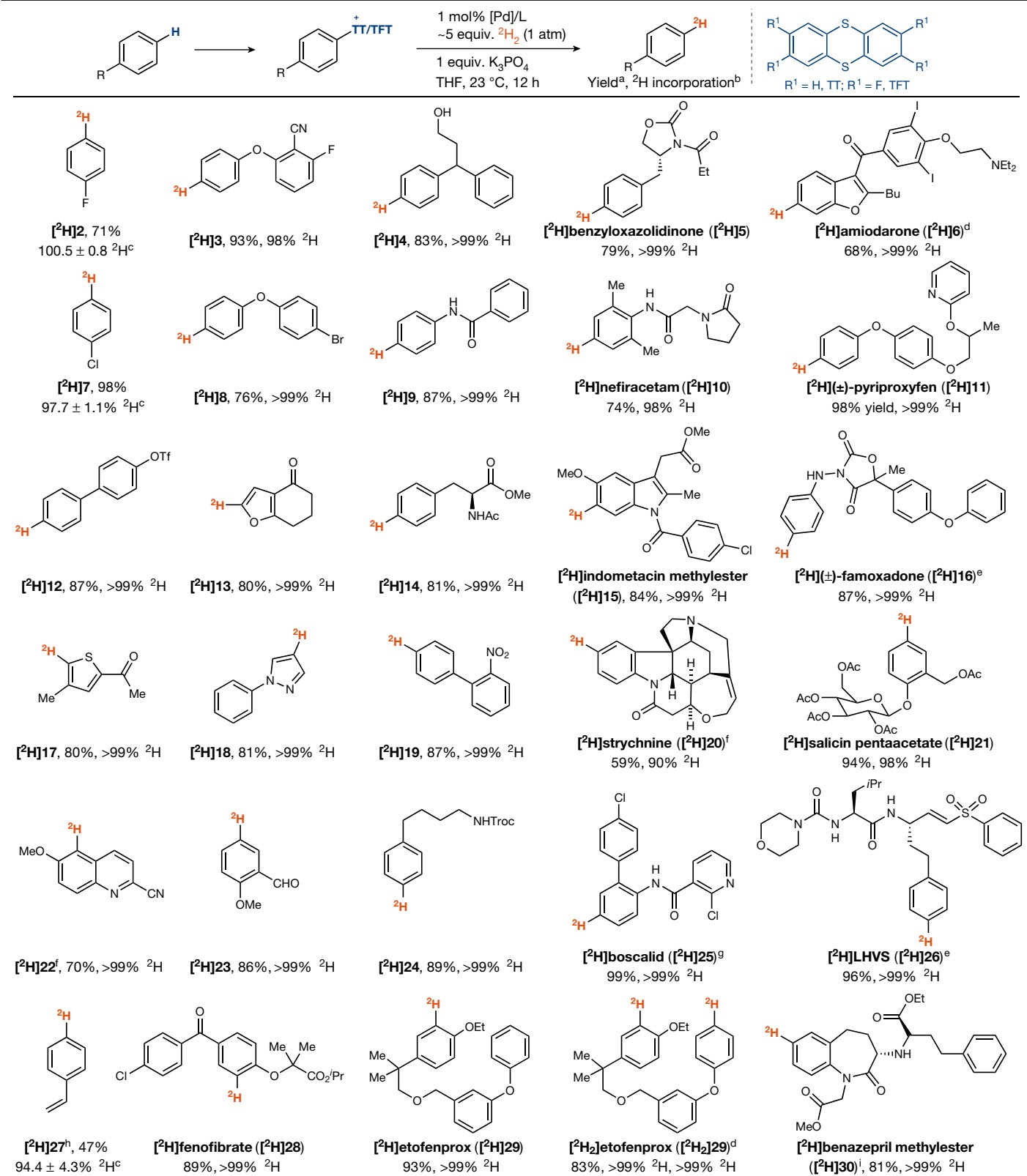

**Fig. 3 | Substrate scope for reductive deuteration of thianthrenium salts.**
Deuterium-labelling of aryl thianthrenium salts. The general reaction
conditions are as follows: (tetrafluoro)thianthrenium salt (0.20 mmol,
1.0 equiv.), K₃PO₄ (1.0 equiv.), Pd(OAc)₂ (1 mol%), dppf (1 mol%) or Pd[(PtBu₃)]₂
1 mol%), THF (0.1 M or 0.2 M), ²H₂ (1 atm, about 5 equiv.), 23 °C, 12 h. ªYield of the
isolated product. ᵇDeuterium incorporation determined by ¹H NMR.
ᶜH incorporation determined by mass analysis. ᵈ5 mol% of catalyst. ᵉ2.5 mol%
of catalyst. ᶠ48 h. ᵍ10 mol% of catalyst. ʰ2 h. ⁱIsolated as triflimide adduct.
TFT, 2,3,7,8-tetrafluorothianthrene.

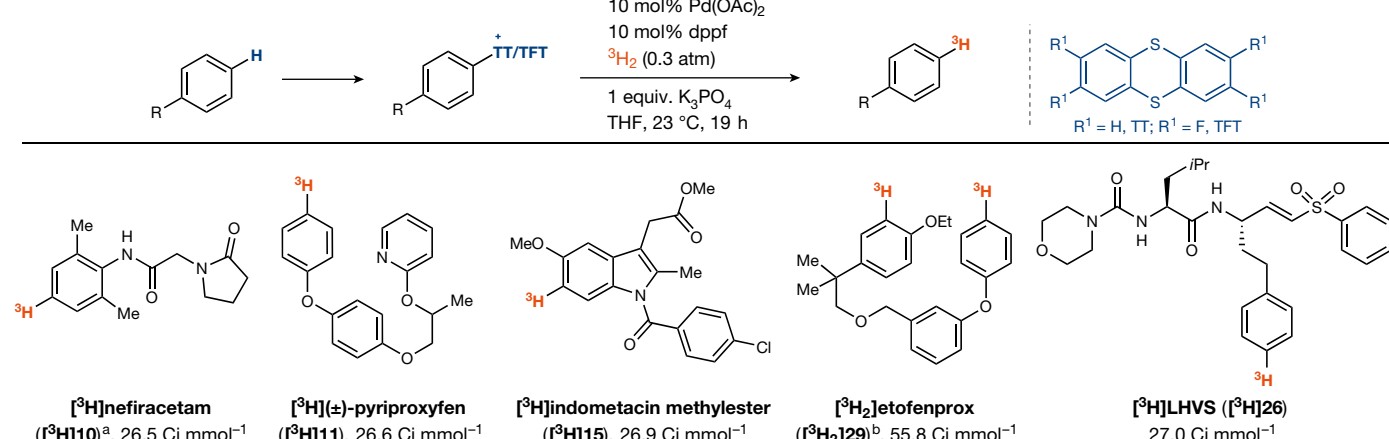

**Fig. 4 | Substrate scope for reductive tritiation of thianthrenium salts.**
Tritium labelling of aryl (tetrafluoro)thianthrenium salts. The general reaction conditions are as follows: (tetrafluoro)thianthrenium salt (3.0 µmol, 1.0 equiv.),

$K_3PO_4$ (1.0 equiv.), Pd(OAc)$_2$ (10 mol%), dppf (10 mol%), THF (6 mM), $^3H_2$ (0.3 atm, about 20 equiv.), 23 °C, 19 h. [a]68 h. [b]$K_3PO_4$ (4 equiv.), $^3H_2$ (0.7 atm, about 20 equiv.), 4 h.

Hydrogenolysis of aryl thianthrenium salts was performed with $^2H_2$ as a more convenient and safer $^3H_2$ surrogate. Palladium-catalysed hydrogenolysis of biphenyl-derived thianthrenium salt **1-TT** with $^2H_2$ afforded the desired $^2H$-labelled product **[²H]1** with more than 99% deuterium (D) incorporation (Fig. 2a). No synthetically useful reaction was observed with aryl bromides, aryl iodides and aryl triflates, nor with other arylsulfonium salts such as those derived from dibenzothiothene (DBT) and diphenyl sulfide (DPS), respectively. The lack of reactivity of the aryl halides could not be addressed through the use of other commonly used, highly active monodentate phosphine ligands (Fig. 2a, Supplementary Table 11). Likewise, the addition of (pseudo)halide anions to the productive reaction of arylthianthrenium salts poisons the reactivity, probably due to palladophilic halide outcompeting dihydrogen for binding; the starting material **1-TT** is recovered in these cases (Fig. 2b). Although sulfur-containing molecules are often responsible for catalyst poisoning owing to strong coordination[41], we could not identify any coordination of thianthrene to cationic Pd(II), which is consistent with an open coordination site on the metal centre for interaction and coordination of $H_2$ during catalysis. As measured by infrared spectroscopy, even triflate outcompetes thianthrene from coordination to Pd(II) (Supplementary Scheme 2, Supplementary Fig. 23). Cationic Pd(II) complexes can commonly be obtained from Pd(II) aryl halides by addition of silver(I) (Ag(I)) salts with non-coordinating anions during catalysis, as exemplified by Overman in Heck reactions that proceed through a mechanistically relevant cationic reaction pathway[42]. However, hydrogenolysis of aryl bromide in the presence of Ag(I) additives did not result in a productive reaction as observed with the arylthianthreniums, which may be due to the incompatibility of oxidizing Ag(I) and the presumed in situ-generated palladium hydride (Fig 2a).

(Tetrafluoro)thianthrenium (TFT and TT, respectively) salts of electron-deficient (**2**, **7**), electron-neutral (**4**, **14**, **24**) and electron-rich arenes (**3**, **8**, **11**), as well as arenes bearing *ortho* substituents (**15**, **17**, **22**) successfully participate in Pd-catalysed reductive deuteration (Fig. 3). Functional-group compatibility is high, as evidenced by substrates containing unprotected hydroxyl groups (**4**), carbamates (**5**), amides (**10**, **20**, **25**), heterocycles (**6**, **13**, **18**) and acid-sensitive glycosidic linkages (**21**). In contrast to heterogeneous catalysis, the homogeneous catalyst system tolerates reducible functional groups such as aryl (pseudo)halides (**2**, **3**, **6**, **7**, **8**), nitriles (**3**), alkenes (**20**, **26**, **27**), esters (**14**, **15**), aldehydes (**23**), sulfones (**26**), nitro groups (**19**) and benzyl ethers (**23**). Primary amines must be protected. No $^1H$–$^2H$ scrambling was observed when the reaction was performed in the presence of water, and high levels of deuterium incorporation were detected throughout.

When compared with heterogeneous deuterodehalogenation, where significant isotopic scrambling can occur[43], the new catalytic platform with aryl thianthrenium salts provides a viable approach for the synthesis of deuterated and tritiated drug molecules (vide infra) with high isotopic purity in a practical setting that does not require the rigorous absence of water or oxygen.

Given our design of electrophilic, cationic Pd(II) species for $H_2$ activation and the observed poisoning of the active catalyst by anions as weakly coordinating as triflate, successful reductive deuteration of substrates featuring coordinating groups such as tertiary amines (**6**, **20**), pyridines (**11**, **25**), pyrazoles (**18**), quinolines (**22**), anilines (**10**, **16**) and thiophenes (**17**) is non-trivial. We observed that the appropriate choice of counterion for the arylthianthrenium salt starting material can reduce its solubility in tetrahydrofuran (THF), with the reservoir of starting material in the solid phase, leaching material for conversion as the reaction progresses. The ability to tune solubility is another yet-unappreciated advantage of the arylthianthrenium salts when compared with arenes with conventional leaving groups that lack such a handle.

Translation of the hydrogenolysis reaction to tritium labelling was accomplished on micromole scales at a subatmospheric pressure of $^3H_2$ gas to reduce the risk of tritium gas leakage and with a higher loading of catalyst to achieve faster reaction rates (Fig. 4). No special care is required to exclude air or moisture during the radiosynthesis, and, in contrast to purification after hydrogen exchange reactions, the radiolabelled product can be readily separated from the starting material owing to the pronounced polarity difference due to the cationic nature of the thianthrenium salts. No isotope scrambling and straightforward purification result in high molar activity, which is often required for receptor binding and occupancy studies[44]. Higher molar activities can be achieved when more than one label is introduced, as shown for $^3H_2$-etofenprox (**[³H₂]29**). In all cases, high, predictable positional selectivity enables single, well defined, labelled molecules that are generally not accessible with hydrogen isotope exchange reactions unless directing groups are used.

A plausible reaction pathway that is consistent with all experimental data is outlined in Fig. 5. The generation of a catalytically active, monoligated Pd(0) catalyst from the only observed resting state Pd[(P$t$Bu$_3$)]$_2$ during catalysis is consistent with the observed reaction order of 0.5 in Pd[(P$t$Bu$_3$)]$_2$. We determined a primary kinetic isotope effect (KIE) of $k_H/k_D$ = 3.1, where $k$ is the rate constant, by measuring the initial rates of independent reactions with $H_2$ and $^2H_2$, respectively, at the same pressure, and an equilibrium isotope effect of 1.1 from an intermolecular

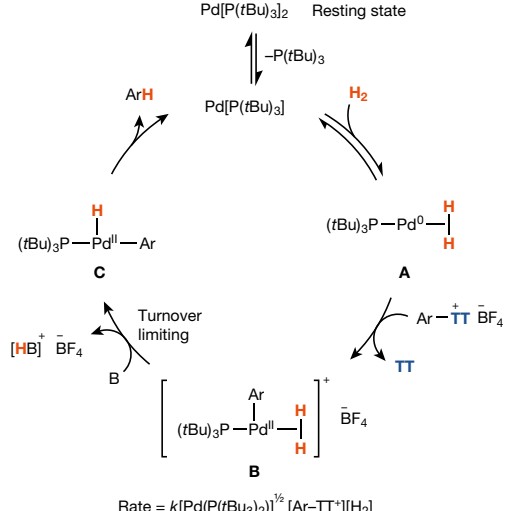

**Fig. 5 | Plausible reaction pathway.** B, K$_3$PO$_4$. Counterion can also be PF$_6$ or NTf$_2$.

competition experiment between H$_2$ and $^2$H$_2$ present at the same partial pressure, which is inconsistent with Ar–TT$^+$ oxidative addition preceding dihydrogen binding but consistent with reversible dihydrogen association before oxidative addition to Ar–TT$^+$. The observed first order in Ar–TT$^+$ and positive order in H$_2$ are in agreement with this proposal. Although we cannot exclude dihydrogen oxidative addition from **A** based on our kinetic data, it would need to be reversible, as indicated by the competition KIE, and the Pd(0) intermediate **A** could be expected to react with Ar–TT$^+$ faster in concerted oxidative-addition or single-electron-transfer pathways than a putative off-cycle Pd(II) dihydride. Irrespective of the exact mechanism of irreversible—measured zero-order in TFT—oxidative addition, which may also differ depending on the ancillary ligand(s), intermediate **B** distinguishes the proposed pathway from those that would be accessible with other aryl(pseudo)halides. No H/D scrambling into the product was observed when hydrogenolysis was performed in the presence of H$_2$O (Supplementary Figs. 4, 5), consistent with irreversible proton transfer from **B**. As the base is not fully dissolved in the liquid phase, its concentration does not appear in the rate law, but the primary KIE excludes turnover-limiting oxidative addition. Similarly, an inverse KIE or a primary KIE much smaller than 3.1 would be observed for turnover-limiting reductive elimination[45] from **C**; moreover, C–H reductive elimination is expected to be fast[46,47]. The data are consistent with turnover-limiting dihydrogen splitting to **C**, followed by fast reductive elimination to product. The addition of halides and pseudohalides or other soluble Lewis bases may prevent formation of **B** in sufficient quantities. Although aryldiazonium salts are typically not accessible through late-stage functionalization, a productive albeit less efficient (Supplementary Scheme 1), hydrogenolysis of biphenyldiazonium tetrafluoroborate under our reaction conditions is consistent with our mechanism hypothesis, as no coordinating anion is generated upon oxidative addition. Although some homogeneous transition-metal complexes can function as precursors for heterogeneous nanoparticles, we have excluded active heterogeneous catalysts as the major contributor to catalysis by means of observing the reaction in the presence and absence of mercury (Supplementary Fig. 22)[48].

In contrast to (pseudo)halides, the thianthrenyl group can be introduced selectively into complex small molecules. We show here how the lack of strong coordination to palladium and the intrinsic solubility properties enables thianthrenium chemistry to solve the challenge of homogeneous palladium-catalysed hydrogenolysis for the chemoselective synthesis of tritium-labelled small molecules.

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

# Article

## Methods

See Supplementary Information for further methods.

## Data availability

The data reported in this paper are available in the main text or the Supplementary Information.

**Acknowledgements** We thank S. Marcus, D. Margold, F. Köhler, N. Haupt and D. Kampen for mass spectrometry analysis; M. Kochius, M. Leutzsch, C. Fares, S. Tobegen and C. Wirtz for NMR spectroscopy analysis; F. Kaegi (F. Hoffmann-La Roche Ltd) and RC Tritec for tritium-labelling experiments; and P. Münstermann for high-performance liquid chromatography analysis. We thank Q. Cheng, B. Lansbergen, F. Juliá and L. Zhang for helpful discussions.

**Author contributions** D.Z. developed the reaction chemistry and investigated the mechanism. D.Z., R.P. and J.Y. optimized and explored the substrate scope for reductive deuteration. D.M. optimized and performed the reductive tritiation reaction. D.Z., D.M. and T.R. wrote the manuscript. T.R. directed the project.

**Funding** Open access funding provided by Max Planck Society.

**Competing interests** T.R. may benefit from royalty payments related to sales from thianthrene-based compounds.

**Additional information**
**Correspondence and requests for materials** should be addressed to Tobias Ritter.
