## [Peer Review File · Nature]

Manuscript Title: Tritiation of aryl thianthrenium salts with a molecular palladium catalyst

Reviewer Comments & Author Rebuttals

Reviewer Reports on the Initial Version:

Referee #1 (Remarks to the Author):

Overall an excellent manuscript and a novel new method for preparing a wide range of specifically deuterated and tritiated pharmaceutical substrates. The one major thing I would like the authors to consider including would be a brief description of the thianthrenium preparation process. While I know the preparation is covered by reference 5 and in the SI, it would be very helpful to the reader if a brief description and summary of the C-H functionalization process were included in the main text before going into the deuterium and tritium labelling experiments, as I suspect that this reaction is not as well known to readers than other more common CH functionalizations. It would then give the reader an opportunity to find all the details for both the preparation of the thianthrenium precursor and the deuteration/tritiation in a single place and make the paper an even more compelling story.

The range of substrates labelled by this method, as is the overall enrichment is impressive. I was curious however if the authors have an explanation for the relatively low deuterium enrichment in Strychnine when compared to all the other substrates.

The detailed optimization and mechanistic studies provided in the SI are also very sound.

I also spotted a few minor errors in the text for correction

Page 2 line 38, 'reaction'

Page 2 line 41, 'application'

Page 2 line 43 and Page 7 line 45, replace 'molar' with 'specific'

Page 3, line 16 'known'

Page 3, line 17, replace 'was' with 'were'

Page 7, line 6 'tritiation'

Referee #2 (Remarks to the Author):

This manuscript by Ritter and co-workers describes the development, application, and investigation of a homogenous hydrogenolysis of aryl thianthrenium salts by a Pd catalyst. The method offers an important synthetic complement to state-of-the-art methods for deuterium/tritium labeling of arenes and heteroarenes, an important reaction in pharmaceutical development for studying ADME of bioactive candidates. First, it uses tritium gas over T2O, which is more toxic and has lower isotopic purity than T2. Second, the method enables site-specific labeling with functional group tolerance and selectivity complementary to isotope exchange reactions, which are most commonly used in these processes. The authors demonstrate application of their method to a broad collection of substrates, including many that are bioactive derivatives. The second highlight of the manuscript pertains to the reactivity and organometallic chemistry. Pd-catalyzed hydrogenolysis of aryl halides with homogenous catalysts has not previously been described due to incompatibility of binding/activation of aryl halides and H2 with the same catalyst. The authors show that aryl thianthrenium salts operate differently and propose that this has to do with the non-coordinating nature of the leaving group as opposed to halides and pseudohalides. A series of reactions designed to evaluate the mechanism of the reaction are presented. Overall, I think the combination of synthetic and mechanistic interest make this

manuscript of potential interest to the readership of Nature. Nevertheless, the mechanistic investigations and discussion in the manuscript require major revision, as indicated below, before the manuscript can be suitable for publication, in my opinion.

1. The comparison of leaving groups in Figure 2a (and in the SI) only surveys conditions that use dppf as a ligand. This ligand is not typically used for cross-coupling of aryl bromides at rt, which calls into question the authors' interpretation of the results as being tied to access to an open coordination site at Pd(II) for H₂ binding. As such, it would be helpful if the researchers compared the reactivity of the leaving groups under conditions more amenable to activation of the aryl halides (electron-rich monodentate phosphines).
2. Figure 2b – what is the mass balance of the reactions in 1, 2, and 3? Do the authors recover aryl thioanthrenium salts or are these converted to aryl bromides/chlorides under the reaction conditions?
3. As mentioned above, the scope of the method appears broad and useful. Are there any particular limitations (beyond modulating the solubility for substrates with Lewis basic groups) that the authors can share? Given that the title is "Tritiation of.." it probably makes sense to show more examples of tritiation in Figure 3 or to adjust the title so that it is a better representation of the research presented in the manuscript.
4. Regarding the mechanism proposed in Figure 4: I think the authors need to do quite a bit more experimentation to elaborate on this proposal and discussion. For example, have the authors considered an alternative cycle wherein H₂ oxidative addition precedes ArX OA? In this scenario, the benefit of using aryl thioanthrenium salts (or diazonium salts) is that they can engage in SET with the Pd(II) dihydride or a Pd(I) intermediate in a radical chain process to generate an aryl radical that recombines with Pd(III)(H) or Pd(II), not that they possess a non-coordinating counterion per se. Stoichiometric organometallic studies, competition experiments for OA of H₂/ArX, combined with kinetic investigation of elementary steps would be able to probe this question alongside in situ interrogation of the catalyst resting state by NMR/ReactIR. Related to this question, I do not understand how the observed 0.5 order dependence on Pd is consistent with the proposed catalytic cycle. All of the on- and off-cycle intermediates in the catalytic cycle of Figure 4 are monomeric. Phosphine dissociation from Pd(PtBu₃)₂ to initiate the catalytic cycle would not lead to ½ order dependence on Pd. And how do the kinetics between PtBu₃-ligated Pd and dppf-ligated Pd compare in this system? More generally, the ligated Pd(II)Ar⁺ doesn't seem particularly electron-deficient, as one might expect for activation of H₂ activation by deprotonation. Ligand effects do not seem consistent with this proposal either.

Referee #3 (Remarks to the Author):

In the present publication, Ritter and coworkers report on the deuteration and tritiation of aryl thianthrenium salts with elemental deuterium and tritium gas using a homogeneous palladium catalyst. Tritiation is an important labeling tool in pharmaceutical chemistry to investigate properties of active compounds such as drugs or receptors. The hitherto available technologies to selectively introduce tritium are limited and only few homogeneous protocols which make use of tritium gas have been developed until to date, which however suffer from functional group tolerance and the sensitivity of the used catalysts. Here, the authors report on a novel protocol which allows for a remarkable group tolerance under mild conditions with an easy-to-use catalyst. The novelty of the protocol clearly lies in the establishment of the first H₂ (or D₂, T₂) activation at a cationic palladium(II) centre, which allows to combine an oxidative addition event with H₂ activation. The trick that the authors applied to realize this combination is the use of thianthrene as leaving group, which due to its poor coordination ability leaves an open active site at palladium for H₂ coordination. The starting reagents, i.e. the thianthrenium salts can be easily synthesized

via C-H functionalization as previously reported by the authors. Overall, I view the reported reaction protocol as a decisive development not only from an application perspective concerning selective tritiation for isotopic labelling, but also from a conceptual/mechanistic point of view. Thus, I rate the manuscript in terms of novelty, significance and clarity as suitable for publication in Nature. Overall, the data provided support the conclusions made in the manuscript. The only exception concerns the proposed mechanism, which in my eyes needs further studies to proof its validity (see below). If the authors can answer these questions I will be happy to accept the manuscript for publication.

- 1) The optimized protocol makes either use of dppf and Pd(OAc)₂ or Pd(PtBu₃)₂. The authors state that these catalysts gave the best yields. According to table S2 other ligands seem to perform superior to PtBu₃. Why was Pd(PtBu₃)₂ selected anyway?
- 2) In their mechanistic studies, the authors used PAdtBu₂ as ligand, which seems to efficiently form cationic Pd complexes. Why was this ligand not included in the ligand scope (Table S2). Does S3 (Scheme S2) react with H₂?
- 3) For two entries in Fig. 3 (2 and 7) a standard deviation is given for the H-incorporation. How was the standard deviation determined?
- 4) Several functional groups are tolerated by the optimized reaction protocol including OH groups. Are primary or secondary amines also tolerated? Since amines are often present in pharmaceuticals, tolerance of this functionality is particularly important and would thus further emphasize the applicability of the method described. In the same vein, is deuteration of the Ar-TT group also selective in presence of not-unactivated alkenes, e.g. in para-TT-styrene?
- 5) The authors conclude from the observed KIE and the lack of isotope wash-in that H₂ activation from B to C is rate-limiting. However, the KIE can also be explained by A ⇌ B being rate-limiting and isotope wash-in with H₂O may also simply be the result of the inability of the cationic complex to be protonated (formation of a dication!). If B ⇌ C (or reductive elimination, see below) is rate-limiting, the reaction rate should depend on the concentration of the base. Is this the case? What is the reaction order in base?
- 6) Complex C in Fig. 4 should be a neutral species. This error must be corrected.
- 7) I don't think that reductive elimination from C is in general a fast process, particularly for small Ar. In this case, reductive elimination might be induced by coordination of a second PtBu₃. Is it possible to detect any intermediate during the reaction process? C (i.e. tBu₃P-Pd(Ar)H or (dppf)Pd(Ar)H) should be accessible from dppf/PtBu₃-Pd(Ph)Br by reaction with e.g. NaBH₄ or LiAlH₄. If reductive elimination is facile, this reaction should instantaneously lead to ArH. The authors should perform these experiments to confirm the proposed mechanism.
- 8) Is formation of B also prevented by addition of phosphine?

Congratulations to these exciting results.

Author Rebuttals to Initial Comments:

Reviewer 1 comments

Reviewer 1, comment 1:

Overall an excellent manuscript and a novel new method for preparing a wide range of specifically deuterated and tritiated pharmaceutical substrates.

Our response:

We are excited about the reviewer's comment.

Reviewer 1, comment 2:

The one major thing I would like the authors to consider including would be a brief description of the thianthrenium preparation process. While I know the preparation is covered by reference 5 and in the SI, it would be very helpful to the reader if a brief description and summary of the C-H functionalization process were included in the main text before going into the deuterium and tritium labelling experiments, as I suspect that this reaction is not as well known to readers than other more common CH functionalizations. It would then give the reader an opportunity to find all the details for both the preparation of the thianthrenium precursor and the deuteration/tritiation in a single place and make the paper an even more compelling story.

Our response:

We thank the reviewer for this comment and agree with the reviewer. Based on the reviewer's comment, we have added a brief description about the C-H thianthrenation step in the main text.

Changes to the manuscript:

On page 3, line 23, we have deleted: "Aryl thianthrenium salts can be more reactive than aryl halides and -triflates in cross-coupling reactions^{5,37} and photoredox catalysis³⁸⁻⁴⁰."

On page 3, line 23, we have added: "Aryl thianthrenium salts can be more reactive than aryl halides and -triflates in cross-coupling reactions^{5,37} and photoredox catalysis³⁸⁻⁴⁰, and they are readily accessed by direct, late-stage C-H functionalization of complex small-molecule arenes in exquisite site-selectivity."

Reviewer 1, comment 3:

The range of substrates labelled by this method, as is the overall enrichment is impressive. I was curious however if the authors have an explanation for the relatively low deuterium enrichment in Strychnine when compared to all the other substrates.

Our response:

We thank the reviewer for this comment. Compared to the other substrates in Figure 3a, the rate of hydrogenolysis of the strychnine-derived tetrafluorothianthrenium salt is lower. In this prolonged reaction time, palladium nanoparticles might be generated via reduction of the homogeneous palladium catalyst by $^2\text{H}_2$ as a side reaction. The strychnine-derived tetrafluorothianthrenium salt can then undergo heterogeneous hydro-defunctionalization with THF solvent as the source of the undesired protium, lowering the ^2H -incorporation in the product to 90% as previously reported. We have performed a control experiment with shorter reaction time under otherwise identical conditions, and the desired [^2H] strychnine was obtained in 95% ^2H -incorporation; we have added this new information and an appropriate discussion to the Supporting Information.

Changes to the SUPPLEMENTARY INFORMATION:

A comment on the preparation of $[{}^2\text{H}]strychnine$ has been added: “Higher ${}^2\text{H}$ -incorporation can be achieved with shorter reaction time (95% deuterium enrichment, 6 h).”

Reviewer 1, comment 4:

The detailed optimization and mechanistic studies provided in the SI are also very sound.

Our response:

We are grateful for the reviewer’s comment.

Reviewer 1, comment 5:

I also spotted a few minor errors in the text for correction

Page 2 line 38, 'reaction'

Page 2 line 41, 'application'

Page 2 line 43 and Page 7 line 45, replace 'molar' with 'specific'

Page 3, line 16 'known'

Page 3, line 17, replace 'was' with 'were'

Page 7, line 6 'tritiation'

Our response:

We thank the reviewer for pointing out these errors in our original manuscript. We have corrected them accordingly.

Reviewer 2 comments

Reviewer 2, comment 1:

This manuscript by Ritter and co-workers describes the development, application, and investigation of a homogenous hydrogenolysis of aryl thianthrenium salts by a Pd catalyst. The method offers an important synthetic complement to state-of-the-art methods for deuterium/tritium labeling of arenes and heteroarenes, an important reaction in pharmaceutical development for studying ADME of bioactive candidates. First, it uses tritium gas over T₂O, which is more toxic and has lower isotopic purity than T₂. Second, the method enables site-specific labeling with functional group tolerance and selectivity complementary to isotope exchange reactions, which are most commonly used in these processes. The authors demonstrate application of their method to a broad collection of substrates, including many that are bioactive derivatives. The second highlight of the manuscript pertains to the reactivity and organometallic chemistry. Pd-catalyzed hydrogenolysis of aryl halides with homogenous catalysts has not previously been described due to incompatibility of binding/activation of aryl halides and H₂ with the same catalyst. The authors show that aryl thianthrenium salts operate differently and propose that this has to do with the non-coordinating nature of the leaving group as opposed to halides and pseudohalides. A series of reactions designed to evaluate the mechanism of the reaction are presented. Overall, I think the combination of synthetic and mechanistic interest make this manuscript of potential interest to the readership of Nature. Nevertheless, the mechanistic investigations and discussion in the manuscript require major revision, as indicated below, before the manuscript can be suitable for publication, in my opinion.

Our response:

We thank the reviewer for the careful analysis of our work.

Reviewer 2, comment 2:

The comparison of leaving groups in Figure 2a (and in the SI) only surveys conditions that use dppf as a ligand. This ligand is not typically used for cross-coupling of aryl bromides at rt, which calls into question the authors' interpretation of the results as being tied to access to an open coordination site at Pd(II) for H₂ binding. As such, it would be helpful if the researchers compared the reactivity of the leaving groups under conditions more amenable to activation of the aryl halides (electron-rich monodentate phosphines).

Our response:

We are grateful for the reviewer's comment and completely agree with the reviewer. While we had initially obtained also other data, with the presentation we ultimately chose for the last submission, we did not address the valid concern the reviewer pointed out. The reason dppf was chosen to be displayed in the manuscript was due to the comparison under otherwise identical conditions of different leaving groups (Figure 2a and Table S9). Based on the reviewers comment, we have now evaluated three additional electron-rich, highly active monodentate phosphine ligands (*t*BuXPhos, BrettPhos and AlPhos) in combination with the aryl (pseudo) halides. The deuterated products were obtained in less than 10% yield in all cases, further supporting our original claims. We added the new data to manuscript and SI, and sincerely thank the reviewer for pointing us in this direction.

Changes to the manuscript:

On page 5, line 14, we have added: “The lack of reactivity of the aryl halides could not be addressed through the use of other commonly used, highly active monodentate phosphine ligands (Fig. 2a, Supplementary Information, Table S11).”

Changes to the SUPPLEMENTARY INFORMATION:

The results of hydrogenolysis of aryl (pseudo) halides with additional electron-rich, monodentate phosphines have been included in the SI (Table S10).

Table S11. Evaluation of additional electron-rich, monodentate phosphine ligands in hydrogenolysis of aryl (pseudo) halides

^aYield was determined by ¹H NMR using mesitylene as an internal standard.

Reviewer 2, comment 3:

Figure 2b – what is the mass balance of the reactions in 1, 2, and 3? Do the authors recover aryl thioanthrenium salts or are these converted to aryl bromides/chlorides under the reaction conditions?

Our response:

Starting material (aryl thianthrenium salt **1-TT**) remained as mass balance for the reactions 1-3 in Figure 2b. The corresponding aryl bromides or chloride is not detected under the reaction conditions; this information is important for the understanding of the story, and we thank the reviewer again for making us aware of our omission; the information was added to the revised manuscript.

Changes to the manuscript:

On page 5, line 18, we have added: "... the starting material **1-TT** is recovered in these cases (Figure 2b)."

Reviewer 2, comment 4:

As mentioned above, the scope of the method appears broad and useful. Are there any particular limitations (beyond modulating the solubility for substrates with lewis basic groups) that the authors can share?

Our response:

We have executed additional experiments with functional groups that were previously not considered. For example, we have added a benzaldehyde and a styrene, further establishing the high functional group tolerance. We have found, however, based on the reviewer's comment, that primary amines are not tolerated. We are pleased to provide an additional example with a secondary amine in the revised substrate scope (**30**), and to also include a primary amine derivative, protected with a Troc group (**24**) both in high yield and isotopic purity. We have specifically pointed out in the revised version that primary amines must be protected.

Changes to the manuscript:

We have added: "Primary amines must be protected."

Reviewer 2, comment 5:

Given that the title is "Tritiation of.." it probably makes sense to show more examples of tritiation in Figure 3 or to adjust the title so that it is a better representation of the research presented in the manuscript.

Our response:

We agree with the reviewer. We are pleased to provide two additional tritiation examples in the manuscript of complex small molecules that have worked as well as the others. The results of the two new tritiation examples have been added to the manuscript and SI.

Changes to the manuscript:

New substrates have been included in the Figure 3b.

Changes to the SUPPLEMENTARY INFORMATION:

The results of new tritiation experiments have been included in the SI.

Reviewer 2, comment 6:

Regarding the mechanism proposed in Figure 4: I think the authors need to do quite a bit more experimentation to elaborate on this proposal and discussion. For example, have the authors considered an alternative cycle wherein H₂ oxidative addition precedes ArX OA? In this scenario, the benefit of using aryl thioanthrenium salts (or diazonium salts) is that they can engage in SET with the Pd(II) dihydride or a Pd(I) intermediate in a radical chain process to generate an aryl radical that recombines with Pd(III)(H) or Pd(II), not that they possess a non-coordinating counterion per se. Stoichiometric organometallic studies, competition experiments for OA of H₂/ArX, combined with kinetic investigation of elementary steps would be able to probe this question alongside in situ interrogation of the catalyst resting state by NMR/ReactIR.

Our response:

We thank the review for the careful analysis. We did not previously consider the mechanistic pathway the reviewer delineates, and are pleased to report additional data that enable us now to make more specific claims. In regards to the stoichiometric studies, we respectfully disagree that these studies can answer the very relevant questions the reviewer poses. Specifically, we have deduced from kinetic measurements that an interaction with H₂ precedes the oxidative addition to arylthianthrenium, however, that interaction is reversible, and, hence, cannot be observed in a stoichiometric reaction. Likewise, while we have evaluated the stoichiometric reaction of the palladium complex with aryl thianthrenium, such a reaction bears no relevance, as it may not be relevant in the presence of dihydrogen, given the faster but reversible reaction with dihydrogen. However, based on the reviewer's comments, we were able to discern through additional kinetic experiments under conditions relevant to catalysis, important conclusions that led to a refined mechanism proposal.

We recorded an equilibrium isotope effect of 1.1 for a competition between H₂ and D₂. In combination with the independent primary KIE, we can exclude dihydrogen binding post Ar-TT+ oxidative addition. Our kinetic data does not allow us to exclude dihydrogen oxidative addition, which we now specifically describe in the manuscript, although it would need to be reversible to fit with the rate data. Rate-limiting oxidative addition to a Pd(II) dihydride is excluded based on the KIE. We have added a revised discussion that also details additional data in the revised manuscript. Reaction with dihydrogen cannot be observed stoichiometrically but such a scenario is also possible for a fast reaction that is accessible but endergonic. While the Pd catalyst reacts quickly with Ar-TT+, it may not do so in the presence of H₂. Although we cannot access the Pd(II) dihydride, the reviewer may agree with us that the reduction potential of the now proposed Pd(0) dihydrogen complex **A** must be higher than the reduction potential of a Pd(II) dihydride complex derived from it, so that

subsequent OA or single electron transfer should be faster from suggested intermediate **A** than from a putative Pd(II) species. Based on the reviewer's comment, we have adopted this discussion in the revised manuscript and sincerely thank the reviewer for challenging us on our previously incomplete mechanism investigation. We now also specifically discuss the eventuality of a Pd(II) dihydride complex.

Reviewer 2, comment 7:

Related to this question, I do not understand how the observed 0.5 order dependence on Pd is consistent with the proposed catalytic cycle. All of the on- and off-cycle intermediates in the catalytic cycle of Figure 4 are monomeric. Phosphine dissociation from Pd(PtBu₃)₂ to initiate the catalytic cycle would not lead to ½ order dependence on Pd.

Our response:

We respectfully disagree with the reviewer. All reactions, in which a ligand has to dissociate from a pre-catalyst that is the resting state, to generate an active catalyst, must have an order of 0.5 in pre-catalyst, as for example in ene-yne metathesis catalyzed by Grubbs catalyst (*Organometallics* **2011**, *30*, 1319) and palladium-catalyzed desulfinate cross-coupling reactions (*J. Am. Chem. Soc.* **2020**, *142*, 3564–3576). The order can be obtained from the math; the relevant point being that the equilibrium constant for ligand dissociation can be used to arrive at the 0.5 order. Intuitively, the order can be understood in the sense that half the material from pre-catalyst proceeds productively, while the other half (dissociated ligand) retards the reaction. We have now added the math in the SI to avoid any confusion. The data supports our proposed mechanism. The analysis is independent of the turnover-limiting step, or any of the following steps, meaning, even if our proposal for the following steps were wrong, we can state unambiguously that ligand dissociation from the pre-catalyst as proposed results in the reaction order of 0.5 for the monomeric palladium pre-catalyst. The rate equations of hydrogenolysis of aryl thianthenium salt based on mechanism shown in Figure 4 are listed as eqs 1-5:

The rate of the hydrogenolysis is shown in eq 6:

$$\text{rate} = k_4 [[\text{P}(\text{tBu})_3]\text{Pd}^+\text{Ar}(\text{H}_2)] [\text{base}] \quad \text{eq 6}$$

By solving the equilibria in eq 3, the concentration of $[\text{P}(\text{tBu})_3]\text{Pd}^+\text{Ar}(\text{H}_2)$ is derived in eq 7:

$$[[\text{P}(\text{tBu})_3]\text{Pd}^+\text{Ar}(\text{H}_2)] = K_3 [[\text{P}(\text{tBu})_3]\text{Pd}^+\text{Ar}] [\text{H}_2] \quad \text{eq 7}$$

Similarly, the concentrations of $[\text{P}(\text{tBu})_3]\text{Pd}^+\text{Ar}$ and $[\text{P}(\text{tBu})_3]\text{Pd}$ can be derived in eq 8-9:

$$[[\text{P}(\text{tBu})_3]\text{Pd}^+\text{Ar}] = K_2 [\text{Pd}[\text{P}(\text{tBu})_3]] [\text{ArTT}^+]/[\text{TT}] \quad \text{eq 8}$$

$$[\text{Pd}[\text{P}(\text{tBu})_3]] = K_1 [\text{Pd}[\text{P}(\text{tBu})_3]_2] / [\text{P}(\text{tBu})_3] \quad \text{eq 9}$$

Considering the concentration of free phosphine equals to that of $[\text{P}(\text{tBu})_3]\text{Pd}$ in eq 1, the concentration of $[\text{P}(\text{tBu})_3]\text{Pd}$ at every point in time is proportional to the square root of the catalyst concentration, as shown in eq 10:

$$[\text{Pd}[\text{P}(\text{tBu})_3]] = (K_1[\text{Pd}[\text{P}(\text{tBu})_3]_2])^{0.5} \quad \text{eq 10}$$

Substituting eqs 7,8 and 10 into eq 6 yields the final rate expression, which is in agreement of the half-order kinetics with respect to palladium catalyst.

$$\text{rate} = k [\text{Pd}(\text{PtBu}_3)_2]^{1/2} [\text{ArTT}^+][\text{H}_2]; \text{ where } k = [K_1]^{1/2} K_2 K_3 k_4 [\text{base}]$$

Changes to the SUPPLEMENTARY INFORMATION:

The analysis of the catalyst rate order has been included in the SI:

Reviewer 2, comment 8:

And how do the kinetics between PtBu₃-ligated Pd and dppf-ligated Pd compare in this system?

Our response:

The reaction with dppf as ligand was performed with Pd(II) (palladium acetate). Analysis of the reaction rate would result in an induction period in the reaction profile for the generation of catalytic active Pd(0) species, thus, a direct comparison between the kinetic profiles between PtBu₃-ligated Pd and dppf-ligated Pd –catalyzed reactions would not enable us to make non-trivial conclusions. We have therefore not performed quantitative kinetic studies to compare the two reaction profiles.

Reviewer 2, comment 9:

More generally, the ligated Pd(II)Ar⁺ doesn't seem particularly electron-deficient, as one might expect for activation of H₂ activation by deprotonation. Ligands effects do not seem consistent with this proposal either.

Our response:

While one might expect such a reactivity, in previous studies with other metals, the correlation the reviewer suggests is not necessarily required. It has been documented that when H₂ binds to a cationic metal center, the acidity of H₂ can be increased by up to 40 orders of magnitude (JACS, **1991**, *113*, 875). Electron-deficient ligands are not necessarily required for such reactivity, for example, a Cp ligand, and phosphine ligands are appropriate for ruthenium. In addition, few palladium species which are supported by electron-rich ligands have been shown to promote heterolytic activation of dihydrogen (*Dalton Trans.*, **2013**, *42*, 6495). While such complexes cannot undergo the reactivity we report here, they, nevertheless, show that electron-poor ligands are not required *a priori* for dihydrogen activation.

Based on the reviewer's comments, we have executed a selection of additional mechanistic experiments. We thank the reviewer for pointing us in this direction. All new data supports the revised, more detailed proposed mechanistic pathway and the claims made in the revised manuscript.

Reviewer 3 comments

Reviewer 3, comment 1:

In the present publication, Ritter and coworkers report on the deuteration and tritiation of aryl thianthrenium salts with elemental deuterium and tritium gas using a homogenous palladium catalyst. Tritiation is an important labeling tool in pharmaceutical chemistry to investigate properties of active compounds such as drugs or receptors. The hitherto available technologies to selectively introduce tritium are limited and only few homogenous protocols which make use of tritium gas have been developed until to date, which however suffer from functional group tolerance and the sensitivity of the used catalysts. Here, the authors report on a novel protocol which allows for a remarkable group tolerance under mild conditions with an easy-to-use catalyst. The novelty of the protocol clearly lies in the establishment of the first H₂ (or D₂, T₂) activation at a cationic palladium(II) centre, which allows to combine an oxidative addition event with H₂ activation. The trick that the authors applied to realize this combination is the use of thianthrene as leaving group, which due to its poor coordination ability leaves an open active site at palladium for H₂ coordination. The starting reagents, i.e. the thianthrenium salts can be easily synthesized via C-H functionalization as previously reported by the authors. Overall, I view the reported reaction protocol as a decisive development not only from an application perspective concerning selective tritiation for isotopic labelling, but also from a conceptual/mechanistic point of view. Thus, I rate the manuscript in terms of novelty, significance and clarity as suitable for publication in Nature. Overall, the data provided support the conclusions made in the manuscript. The only exception concerns the proposed mechanism, which in my eyes needs further studies to proof its validity (see below). If the authors can answer these questions I will be happy to accept the manuscript for publication.

Our response:

We thank the reviewer for their detailed analysis of our work.

Reviewer 3, comment 2:

The optimized protocol makes either use of dppf and Pd(OAc)₂ or Pd(PtBu₃)₂. The authors state that these catalysts gave the best yields. According to table S2 other ligands seem to perform superior to PtBu₃. Why was Pd(PtBu₃)₂ selected anyway?

Our response:

We thank the reviewer for this comment and regret that we did not explain our choice in the original submission. Other ligands perform faster than PtBu₃ in the reaction of pyriproxyfen-derived thianthrenium salt. However, we have found that, for substrates containing strong coordinating groups, Pd[(PtBu₃)₂] displays superior catalytic performance (tetrafluoroborate was generally the counterion of choice in these cases) whereas the best conditions from Table S2 failed to deliver any

products. The observed higher activity of bulky and electron-rich ligand PtBu_3 in these cases may be related to avoiding the coordination of substrate to the T-shaped cationic arylpalladium(II) complex (JACS, **2004**, 126, 1184; Dalton Trans., **2014**, 43, 13545). We have provided additional data in the Supporting Information that make the choice of ligand more obvious, and thank the reviewer for pointing us in this direction. In the revised manuscript, we also specifically point out why PtBu_3 is the preferred ligand for such cases.

Changes to the SUPPLEMENTARY INFORMATION:

A table of selected examples for which $\text{Pd}[(\text{PtBu}_3)]_2$ displays superior catalytic performance has been included in the SI (Table S9).

For the following substrates, $\text{Pd}[(\text{PtBu}_3)]_2$ displays superior catalytic performance over $\text{Pd}(\text{OAc})_2$ and dppf .

Table S9. Superior catalytic performance of $\text{Pd}[(\text{PtBu}_3)]_2$ in the preparation of deuterated famoxadone and boscalid.

Reviewer 3, comment 3:

In their mechanistic studies, the authors used PAdtBu_2 as ligand, which seems to efficiently form cationic Pd complexes. Why was this ligand not included in the ligand scope (Table S2). Does S3 (Scheme S2) react with H_2 ?

Our response:

The reason this ligand was not evaluated in Table S2 is because it is synthetically much less accessible. It must be prepared by a 4-step inert atmosphere synthesis. We had only included commercially available ligands for practicality. Scientifically, however, we completely agree with the reviewer. Therefore, as suggested by the reviewer, we have evaluated the PAdtBu₂ ligand, and the desired deuterated product was obtained in 92% isolated yield, similarly to what we observe with the commercially available Pd[(P^tBu₃)₂]. The reaction of T-shaped aryl Pd(II) triflate **S3** with ²H₂ was also performed and 0% of ²H-incorporation was detected in the product (benzene), consistent with our hypothesis. These new results have been added to the SI. While of little practical value, the results are fully consistent with the proposed mechanism and better complete the story; we thank the reviewer for the suggestion to evaluate that ligand in the reaction.

Changes to the SUPPLEMENTARY INFORMATION:

The results of PAdtBu₂ have been added to Table S2, entry 16;

The results from the reaction of aryl Pd(II) triflate **S3** with H₂ have been added to the SI (Scheme S3).

Scheme S3. Reaction of T-shaped Pd(II) triflate **S-3** with ²H₂.

The T-shaped aryl Pd(II) triflate **S-3** is not stable, it decomposes rapidly to a complex mixture from which 0% ²H incorporation was detected in benzene by mass analysis.

Reviewer 3, comment 4:

For two entries in Fig. 3 (2 and 7) a standard deviation is given for the H-incorporation. How was the standard deviation determined?

Our response:

We thank the reviewer for this comment and regret that we have not made clear enough how we determined the standard deviation. For these few compounds, due to volatility, we determined the ²H-incorporation by mass analysis. The software package used to perform the analysis is MassLib (V9.4, MSP Kofel, Zollikofen, Switzerland). The signal group of the molecular ion of the (possibly) deuterated compounds is matched against clean spectra obtained from the non-labelled reference (²H₀) and copies of this spectrum, shifted by 1 times the corresponding amount of ²H (e.g. +2 for ²H₂).

The applied algorithm then tries to find the best matching combination of these theoretical spectra against the experimental one to calculate the ^2H -incorporation and the corresponding standard deviation. We have this now mentioned in the revised manuscript, and explained in the revised SI.

Changes to the manuscript:

A figure caption has been added to Figure 3a:

^2H -incorporation was determined by mass analysis.

Changes to the SUPPLEMENTARY INFORMATION:

Details of determination of ^2H -incorporation is added to the general information of SI;

For volatile deuterated substrates, ^2H -incorporation was determined by mass analysis:

The software package used to perform the analysis is MassLib (V9.4, MSP Kofel, Zollikofen, Switzerland). The signal group of the molecular ion of the (possibly) deuterated compounds is matched against clean spectra obtained from the non-labelled reference ($^2\text{H}_0$) and copies of this spectrum, shifted by 1 times the corresponding amount of ^2H (e.g. +2 for $^2\text{H}_2$). The applied algorithm then tries to find the best matching combination of these theoretical spectra against the experimental one to calculate the ^2H -incorporation and the corresponding standard deviation.

Reviewer 3, comment 5:

Several functional groups are tolerated by the optimized reaction protocol including OH groups. Are primary or secondary amines also tolerated? Since amines are often present in pharmaceuticals, tolerance of this functionality is particularly important and would thus further emphasize the applicability of the method described. In the same vein, is deuteration of the Ar-TT group also selective in presence of not-unactivated alkenes, e.g. in para-TT-styrene?

Our response:

We have found that primary amines are not tolerated and must be protected. However, we are pleased to report additional data to show a protected amine in the revised substrate table (24).

In addition, based on the reviewer's comment, we have added a secondary amine to the substrate table.

Activated alkenes, such as in a styryl TT salt the reviewer suggested can be tolerated in our reaction, as demonstrated by the hydrogenolysis of the 4-vinylbenzene-derived thianthrenium salt. We have also added this new data to manuscript and SI:

Changes to the manuscript:

The new substrates have been added to Figure 3a.

Changes to the SUPPLEMENTARY INFORMATION:

The results from the reaction of new substrates have been added to the SI.

Reviewer 3, comment 6:

The authors conclude from the observed KIE and the lack of isotope wash-in that H₂ activation from B to C is rate-limiting. However, the KIE can also be explained by A to B being rate-limiting and isotope wash-in with H₂O may also simply be the result of the inability of the cationic complex to be protonated (formation of a dication!). If B to C (or reductive elimination, see below) is rate-limiting, the reaction rate should depend on the concentration of the base. Is this the case? What is the reaction order in base?

Our response:

We thank the reviewer for the careful analysis. We are pleased to provide additional experimental data to gain more insights into the reaction mechanism. Specifically, we have observed an equilibrium KIE in a competition experiment. The reaction order in base cannot be determined as it is in a different phase, by design, an insoluble base must be used. The use of soluble bases, even hindered ones such as 2,6-ditert-butyl pyridine prevent the reaction. We are pleased to provide additional data that exclude oxidative addition as rate-limiting. The concentration of base, therefore, is not part of the rate equation, which we now specifically mention in the revised manuscript.

Inverse kinetic isotope effects are usually observed for the formation of dihydrogen transition metal complexes (JACS, **1997**, 119, 9179; JACS, **2009**, 131, 11263; JACS, **1999**, 121, 11402). We determined an equilibrium isotope effect of 1.1 from intermolecular H₂/D₂ competition and a primary

KIE of 3.1 from independent rate constants for reaction with H₂ and with D₂. This data excludes rate-limiting oxidative addition.

parallel reactions: $k_H/k_D = 3.1$

intermolecular competition: $k_H/k_D = 1.1$

Changes to the manuscript:

“In addition, we determined a primary kinetic isotope effect (KIE) of $k_H/k_D = 3.1$ by measuring the initial rates of the reaction with H₂ and ²H₂ at the same pressure and a primary KIE of 1.1 from an intermolecular competition between H₂ and ²H₂ at the same partial pressure.”

Changes to the SUPPLEMENTARY INFORMATION:

The results from the pressure dependence in hydrogenolysis and intermolecular competition experiments between H₂ and ²H₂ have been added to the SI.

Reviewer 3, comment 7:

Complex C in Fig. 4 should be a neutral species. This error must be corrected.

Our response:

We thank the reviewer for pointing out this error in our original manuscript. We have corrected the structure of complex C in Figure 4.

Changes to the manuscript:

The structure of complex **C** in Figure 4 have been corrected.

Reviewer 3, comment 8:

I don't think that reductive elimination from **C** is in general a fast process, particularly for small Ar. In this case, reductive elimination might be induced by coordination of a second PtBu₃. Is it possible to detect any intermediate during the reaction process? **C** (i.e. tBu₃P-Pd(Ar)H or (dppf)Pd(Ar)H) should be accessible from dppf/PtBu₃-Pd(Ph)Br by reaction with e.g. NaBH₄ or LiAlH₄. If reductive elimination is facile, this reaction should instantaneously lead to ArH. The authors should perform these experiments to confirm the proposed mechanism.

Our response:

Only the PdL₂ can be observed as resting state. We respectfully disagree with the reviewer in the sense that C-H reductive elimination may be slow. Theoretical studies by Goddard (JACS. **1984**, *106*, 8321; JACS, **1986**, *108*, 6115), and Whitesides (*Inorg. Chem.*, **1982**, *21*, 2162) suggest that C-H reductive elimination from Pd(II) is facile and extremely fast for Pd(II) species (e.g., the activation barrier for reductive C-H coupling from Pd(II)(H)(Me) is about 10.4 kcal/mol). Hartwig reported that, when compared to four-coordinate aryl Pd(II) species, rates of reductive elimination from three-coordinate aryl palladium(II) species such as **C** are even faster without association or reversible dissociation of the phosphine ligand (JACS. **2004**, *126*, 5344). In addition, in contrast to other Pd(II) aryl X compounds, no Pd(II) aryl hydride could ever be isolated (with the aryl not being tied up by other ligands), consistent with extremely fast reductive elimination.

The proposed aryl palladium hydride intermediate has not been accessed by the reaction of the corresponding aryl palladium halide with hydride source, as it will lead to the reduction of the Pd(II) to Pd(0) species, possibly by extremely fast reductive elimination. We have monitored the reaction by low temperature NMR and the only observed palladium species is Pd[(PtBu₃)₂]. We have added this information to the revised manuscript. In addition, the observed primary KIE is not consistent with rate-limiting reductive elimination, as one would expect an inverse KIE or a small primary one. We have added a revised discussion with references in the revised manuscript, and thank the reviewer for challenging us to better explain this aspect in the revised manuscript.

Reviewer 3, comment 9:

Is formation of **B** also prevented by addition of phosphine?

Our response:

As suggested by the reviewer, we have performed two control experiments in which 1.0 equivalent of the phosphine ligand was added. In this case, the formation of the deuterated product were dramatically suppressed, which suggests the formation of intermediate **B** can be prevented by

excess phosphine. Likewise, the reaction order of 0.5 of the palladium catalyst also establishes that addition of additional ligand must reduce the efficiency of the reaction.

Changes to the SUPPLEMENTARY INFORMATION:

The effect of excess phosphine ligand in the hydrogenolysis reaction has been included in the SI as Table S10.

Table S10. Effect of excess phosphine ligand in hydrogenolysis.

Reviewer 3, comment 10:

Congratulations to these exciting results.

Our response:

We are grateful for the reviewer's comment.

Reviewer Reports on the First Revision:

Referee #2 (Remarks to the Author):

The authors have addressed all of my questions and concerns. I have no reservations supporting this revised manuscript for publication.

Referee #3 (Remarks to the Author):

Ritter and coworkers provide a revised version of their paper on the Pd-catalyzed tritiation of aryl thianthrenium salts, in which they have addressed all question I raised based on the original submission. The main reservations I had concerned the proposed mechanism, in particular that alternative pathways haven't been considered. The authors have now conducted a series of further experiments that confirm their original hypothesis. The reasoning seems to be conclusive and coherent. Furthermore, they have added additional details on the substrate scope and on the optimization of the reaction protocol. In my eyes, the authors have provided sufficient proof for the conclusions made. I cannot find any further inconsistencies or shortcomings. Thus, I recommend publication of this fine work in Nature.